# Trajectories of Lipid Profile and Risk of Carotid Atherosclerosis Progression: A Longitudinal Cohort Study

**DOI:** 10.3390/nu14153243

**Published:** 2022-08-08

**Authors:** Haixu Yu, Yanguang Li, Liyuan Tao, Lincheng Yang, Dan Liu, Yang Wang, Xiaoyan Hao, Honghai He, Ying Che, Peng Wang, Wei Zhao, Wei Gao

**Affiliations:** 1NHC Key Laboratory of Cardiovascular Molecular Biology and Regulatory Peptides, Key Laboratory of Molecular Cardiovascular Science, Ministry of Education, Beijing Key Laboratory of Cardiovascular Receptors Research, Department of Cardiology and Institute of Vascular Medicine, Peking University Third Hospital, Beijing 100191, China; 2Research Center of Clinical Epidemiology, Peking University Third Hospital, Beijing 100191, China; 3Physical Examination Center, Peking University Third Hospital, Beijing 100191, China

**Keywords:** lipid profile, longitudinal cohort, carotid atherosclerosis, trajectory

## Abstract

Background: Early assessment of carotid atherosclerotic plaque characteristics is essential for atherosclerotic cardiovascular disease (ASCVD) risk stratification and prediction. We aimed to identify different trajectories of lipid profiles and investigate the association of lipid trajectories with carotid atherosclerosis (CAS) progression in a large, longitudinal cohort of the Chinese population. Methods: 10,412 participants aged ≥18 years with ≥2 times general health checkups were included in this longitudinally prospective cohort study at Peking University Third Hospital. We used latent class trajectory models to identify trajectories of total cholesterol (TC), low-density lipoprotein cholesterol (LDL-C), triglycerides (TG), and high-density lipoprotein cholesterol (HDL-C) over follow-up time (757 days, IQR: 388–844 days). Results: Participants with carotid plaque were more likely to be older, male, have higher body mass index, have a higher prevalence of hypertension and diabetes, and have a higher level of blood pressure, TG, TC, and LDL-C, compared with carotid intima-media thickness (cIMT) and normal group. Subjects were trichotomized according to different trajectory patterns into stable, moderate-stable, and elevated-increasing classes. TC ≥ 5.18 mmol/L and moderate-stable class (hazard ratio (HR): 1.416, 95% confidence interval (CI): 1.285–1.559, *p*: 0.000), TG ≥ 1.70 mmol/L and moderate-stable class (HR: 1.492, 95% CI: 1.163–1.913, *p*: 0.002), TG ≥ 1.70 mmol/L and elevated-increasing class (HR: 1.218, 95% CI: 1.094–1.357, *p*: 0.000), LDL-C ≥ 3.36 mmol/L and stable class (HR: 1.500, 95% CI: 1.361–1.653, *p*: 0.000) were statistically significant associated with CAS progression compared with the reference group. Conclusions: Borderline elevated baseline lipid (TC, TG, and LDL-C) with stable and elevated-increasing trajectories were associated with CAS progression. Long-term strategies for low-level lipid are beneficial for ASCVD management.

## 1. Introduction

Cardiovascular diseases (CVDs), including coronary heart disease, cerebrovascular disease, and peripheral arterial disease, remain the leading cause of morbidity and mortality, raising a tremendous health burden worldwide [1]. It is estimated that 17.9 million people died from CVDs in 2019, accounting for 32% of all global deaths, 85% of which were due to heart attack and stroke. Over three-quarters of CVD deaths occurred in low- and middle-income countries [1]. In addition, CVDs still occupied the leading cause of death in China, the world’s most considerable low- and middle-income country. According to the annually cardiovascular report published in 2019, the prevalence and mortality of CVD in China are persistently rising, which have become the most troublesome leading public health issue [2]. The most effective strategy is to control the critical pathophysiological process in CVDs early before the onset and progression.

As the primary pathophysiological process, atherosclerosis is responsible for multiple CVDs that start early in life with a long subclinical phase before the clinical manifestation of disease and symptoms that vary depending on the presence and degree of underlying plaque [3]. Therefore, early assessment of atherosclerotic plaque characteristics is essential for risk stratification and prediction of atherosclerotic cardiovascular disease (ASCVD). Carotid ultrasonography is a rapid, noninvasive, and repeatable imaging method to evaluate carotid atherosclerosis (CAS) and underlying cardiovascular risk. Beyond overall risk stratification, CAS is the most common cause of ischemic stroke. About 18−25% of total strokes are caused by the result of CAS [4,5]. Two distinct approaches have been used to assess CAS: The measurement of carotid intima-media thickness (cIMT) and the assessment of carotid arterial plaque. A meta-analysis of large-scale patients has demonstrated that cIMT progression was a widely useful surrogate marker for CVD risk [6]. The ASCVD risk reduction is dependent on the extent of the therapeutical effect on cIMT progression. In addition, carotid artery plaque, which is estimated by carotid ultrasonography as a possible reclassified CVD risk, may be considered a risk modifier in patients at intermediate ASCVD risk [7,8].

As the essential pathogenesis, the unfavorable lipid profile has been understood in the long-term as a causative risk factor in the development and progression of ASCVD. Dyslipidemia comprises a range of conditions, mainly defined by elevations in lipoprotein cholesterol, including total cholesterol (TC), low-density lipoprotein cholesterol (LDL-C), as well as elevated triglycerides (TG), and declined high-density lipoprotein cholesterol (HDL-C) [9]. These different changes in specific lipid composition are involved in the various progression of atherosclerosis. Large-scale evidence revealed a compelling association between dyslipidemia and a higher risk of ASCVD, which has long been recommended and applied by guidelines in clinical practice [9,10]. However, risk factors are dynamically varied, and the risk of constant dyslipidemia probably contributed to the higher risk of atherosclerosis progression. Clinicians should also pay more attention to consistently high levels of lipid profile rather than single or occasional elevations. Several publications suggested that repeating risk assessments can modestly improve prediction compared with a single risk assessment [11]. On the other hand, the evaluation of the trajectory of lipid profile is becoming particularly significant. Previous publications reported the association between different lipid trajectories and incident cardiovascular disease [12,13]. The underexplored issue is whether the overall lipid profiles (i.e., trajectory) are associated with CAS progression over time. Therefore, our study aimed to identify different trajectories of lipid profiles and to investigate the association of lipid trajectories with CAS progression in a large, longitudinal cohort of the Chinese population.

## 2. Materials and Methods

### 2.1. Study Design and Setting

This longitudinally prospective cohort study was collected from population-based general health checkups at Peking University Third Hospital from January 2011 to December 2020. Initially, 76,683 participants older than 18 years who underwent general medical examinations, were included. Major exclusion criteria include those with ≤1 time of carotid ultrasonography. A total of 10,412 participants were enrolled in this study ultimately. The research flowchart is shown in Figure 1. This study was conducted in accordance with the Declaration of Helsinki and approved by the Peking University Third Hospital Medical Science Research Ethics Committee (M2021098). All patients provided written informed consent. This study followed the Strengthening the Reporting of Observational Studies in Epidemiology (STROBE) reporting guideline for cohort studies [14].

### 2.2. Characteristics and Measurement

Demographic characteristics and medical history were collected from the questionnaire. Systolic blood pressure (SBP, mmHg) and diastolic blood pressure (DBP, mmHg) were recorded as the average of three measurements on the right upper arm in the sitting position after 5-min rest. Height (in meters) and weight (in kilograms) were taken according to the standard protocols with shoes removed and the participants wearing light clothing. Body mass index (BMI) was calculated as weight (in kilograms) divided by the square of height (in meters). Hypertension was defined as systolic blood pressures ≥ 140 mmHg and/or diastolic blood pressures ≥ 90 mmHg or the current use of antihypertensive medications. Diabetes was defined as fasting blood glucose (FBG) ≥7.0 mmol/L in the cohort exam or self-reporting a history of diabetes diagnosed by physicians. Peripheral blood samples were tested by the Clinical Laboratory Department of Peking University Third Hospital with a laboratory accreditation certificate. Biochemical parameters, including FBG, TC, TG, LDL-C, and HDL-C, were determined with a commercially available assay kit on the Beckman Coulter platform (Beckman Coulter Inc., Brea, CA, USA) clinical chemistry analyzer.

### 2.3. Carotid Ultrasonography Examination

The ultrasonographic assessments of bilateral carotid arteries (common, bifurcation, internal and external carotid arteries) were performed manually by experienced and certified doctors from the Ultrasound Department from Peking University Third Hospital, who were blinded to this study at baseline and follow-ups, using B-mode, color-mode, and Doppler mode with GE^®^ Vivid i/E95 high-resolution ultrasound system (GE Healthcare, Milwaukee, WI, USA) and 7.5–12 MHz phased array probe. The region of interest for cIMT came from lateral longitudinal projection and the far wall of the bilateral carotid arteries (mid and distal common carotid artery, 1 cm proximal to the carotid bulb) and the carotid bifurcation (1 cm proximal to the flow divider). Abnormal cIMT was defined as the maximum cIMT value ≥0.9 mm, which was the maximum distance between the interface of the lumen-intima and media-adventitia. The carotid plaque was defined as a focal structure protruding into the arterial lumen of at least 0.5 mm or 50% of the surrounding cIMT value or showing a thickness >1.5 mm measured from the media-adventitia interface to the intima-lumen interface [8,15,16]. Carotid atherosclerosis (CAS) progression was defined as the appearance of newly developed carotid stenosis, CAS plaque, and cIMT during follow-up compared with the baseline. For individuals with combined CAS plaque and cIMT, baseline data and follow-up outcomes were defined according to superior manifestation (i.e., CAS plaque) [17,18].

### 2.4. Statistical Analysis

Considering skewed distribution, continuous variables were expressed as medians with interquartile range (IQR). Categorical variables were expressed as absolute numbers with percentages (%). Comparisons of continuous variables were analyzed using Mann-Whitney *U* tests or Kruskal-Wallis *H*-test (two or more independent samples), and comparisons of categorical variables were analyzed using the Chi-squared test or Fisher’s exact tests. We calculated hazard ratio (HR) with the 95% confidence interval (CI) using the Cox proportional hazards regression model to investigate the associations among lipid profiles for CAS progression.

Latent class trajectory modeling (LCTM) is a relatively new methodology in epidemiology to describe life-course exposures. In this study, we used LCTM to characterize the trajectories of four lipid profiles (TG, TC, LDL-C, and HDL-C) over follow-up time. This specialized form of finite mixture modeling allowed us to simplify the heterogeneously longitudinal course of lipid profiles into homogeneous classes and investigate latent classes of participants following similar trajectories over time [19]. Models were fitted using the *lcmm* package (version 1.9.5) in R (version 4.2.0, Vienna, Austria). The best-fitting number of classes chosen was based on the following criteria: (1) The lowest and Bayesian information criteria (BIC) while maintaining clinical meaningfulness and model parsimony; (2) the average probabilities of assignments above 70% for all latent classes; and (3) proportion of individuals estimated to be assigned to each class size were above 2% of the population [19]. Meanwhile, clinical characterization and plausibility of trajectory classes were considered. Finally, three distinct trajectories of each lipid profile were selected as the best-fitting model. To facilitate interpretability, we assigned labels to the trajectories based on their visualized patterns and clinical characteristics.

Trajectory group characteristics were compared, as appropriate, via ANOVA or Kruskal-Wallis H-test for continuous variables and Chi-squared test for categorical variables. Cox proportional hazards regression model with follow-up time as the time scale for trajectories was used to investigate the association between trajectory classes and CAS progression. Moreover, we considered multiple potential confounders based on this study. Then, we included age, male, BMI, hypertension, and diabetes as the covariate. According to recommendations or consensus on dyslipidemia and cardiovascular disease prevention, baseline lipid levels were defined as borderline criteria (TC ≥ 200 mg/dL, TG ≥ 150 mg/dL, LDL-C ≥ 130 mg/dL, HDL-C < 40 mg/dL) [9,20]. Lipid values were converted to SI units. Combined with baseline lipid levels and trajectories, a forest plot was constructed using the *ggcorrplot* package (version 0.1.3). To assess the robustness of our main findings, we performed two sensitivity analyses by restricting the analysis to participants without chronic diseases (hypertension and diabetes), and the young population (age ≤ 65 years old) to reduce the possibility of association caused by chronic diseases and aging.

All statistical analyses were performed by IBM SPSS software (version 23.0, SPSS Inc., Chicago, IL, USA), GraphPad Prism (version 7.0, La Jolla, CA, USA), and R (version 4.2.0) with RStudio (version 2022.02.3, Boston, MA, USA) and associated packages. Two-tailed with *p*-values < 0.05 were considered statistically significant. The Bonferroni correction was also considered for the multiple group comparison (*p* < 0.017).

## 3. Results

### 3.1. Characteristics of Participants

Among the total of 76,683 participants in screening, 10,412 participants (52.1% men) ultimately entered this study (Figure 1); the median age at enrollment was 50 years (IQR: 40–61), with the median BMI of 24.42 kg/m^2^ (IQR: 22.21–26.71). During the median follow-up of 757 days (IQR: 388–844 days), 1815 (17.4%) reached CAS progression. Baseline characteristics of the demographic information, medical history obtained from the questionnaire, and biochemical parameters of study participants were shown in Table 1. Participants with carotid artery plaque were more likely to be older, men, have higher BMI, have a higher prevalence of hypertension and diabetes, have a higher level of blood pressure, FBG, TC, TG, HDL-C, and LDL-C, compared with cIMT group or normal group (all *p* for trend < 0.01).

### 3.2. Characterization of Trajectories of Lipid Profiles

According to model-adequacy criteria and the rule of interpretability, various trajectories of the four lipid profiles were subsequently selected. According to the criteria of the best-fitting number of classes, Figure 2 showed the final latent class trajectory models. Three distinct trajectories were identified for each lipid ingredient. TC, LDL-C, and HDL-C trajectories were similarly demonstrated as U-shape, inverse U-shape, and stable state. The trajectories of TG were shown as the elevated-increasing class, moderate-stable class, and low-stable class.

The baseline demographics and clinical characteristics of lipid trajectories defined by LCTM were shown in Table 2. Participants with inverse U-shape or U-shape trajectory of TC were more likely to be aging, female, more likely to have hypertension, diabetes, a higher level of BMI, blood pressure, baseline lipid profiles (TC, TG, LDL-C), or FBG. The overall baseline characteristics of LDL-C were similar to those of TC. Compared with the stable group of HDL-C, participants with inverse U-shape or U-shape trajectory of HDL-C tended to be older, female, lighter weight, less likely to have hypertension and diabetes, and lower baseline levels of TG, LDL-C, and FBG. Compared with the low-stable class of TG, elevated-increasing and moderate-stable trajectories of TG have more comprehensive CVD risk factors, including age, men, BMI, hypertension, diabetes, blood pressure, baseline lipid profiles, and FBG.

### 3.3. Association of Lipid Trajectories with CAS Progression

The relationships between baseline lipids and CAS progression were presented in Table 3. Considering clinically borderline-high levels, unadjusted baseline lipid profiles (TC ≥ 5.18 mmol/L, TG ≥ 1.70 mmol/L, LDL-C ≥ 3.36 mmol/L) were independently associated with CAS progression. After adjusting for age, male, BMI, hypertension, and diabetes covariates, the baseline borderline-high level of TC and LDL-C had 1.316-fold (95% CI: 1.194–1.451) and 1.382-fold (95% CI: 1.254–1.523) risk of CAS progression. As for the associations between lipid trajectories and CAS progression (Table 3), only two trajectories of TG showed significance. The unadjusted HRs were 1.497 (95% CI: 1.174–1.908) and 1.192 (95% CI: 1.078–1.318) for the elevated-increasing class and moderate-stable class, respectively. The elevated-increasing class had a 1.417-fold (95% CI: 1.101–1.824) risk of CAS progression.

We considered combining baseline lipid profiles and trajectories to explore the relationship between participants’ lipid and CAS progression. The Forest plot showed the association between CAS progression and different subgroups (Figure 3). Participants with baseline TC ≥ 5.18 mmol/L and moderate-stable class had a 1.416-fold (95% CI: 1.285–1.559) risk of CAS progression. Participants with baseline TG ≥ 1.70 mmol/L and elevated-increasing class and TG ≥ 1.70 mmol/L and moderate-stable class had 1.218-fold (95% CI: 1.094–1.357) and 1.492-fold (95% CI: 1.163–1.913) risk of CAS progression. Participants with baseline LDL-C ≥ 3.36 mmol/L and stable class had a 1.500-fold (95% CI: 1.361–1.653) risk of CAS progression.

### 3.4. Sensitivity Analysis

Appendix A demonstrated detailed results for the sensitivity analysis. After restricting the analysis to participants without the chronic disease (hypertension and diabetes), and the young population (age ≤ 65 years old), the subsequent risks of CAS progression by lipid trajectory classes from baseline also yielded similar results as the main analyses. According to WHO criteria, we also stratified the age levels in an association of lipid trajectories with CAS progression (shown in Appendix A). The significance for unadjusted TG trajectories showed mainly differences among the young population (age ≤45 years). In the unadjusted TC trajectories, CAS progression was significantly associated with U-shape TC trajectory among the young population. In comparison between young, middle-aged, and elderly populations, lipid trajectories differed and warranted further investigation.

## 4. Discussion

In this large-scale longitudinal cohort study, to our knowledge, we first described lipid trajectories using LCTM to evaluate the associations between baseline and longitudinal change in lipid profile and the risk of CAS progression. Our results suggested that borderline-elevated baseline lipid levels (TC, TG, and LDL-C) with stable/increasing trajectories were significantly associated with a higher risk of CAS progression. The prognostic value of longitudinal TG trajectories in the CAS progression was independent of baseline TG levels. Without assessing baseline cholesterol levels, there was no prominent relationship between cholesterol trajectories (TC, LDL-C, and HDL-C) and CAS progression.

### 4.1. Independent and Joint Effects of Baseline and Trajectories of Lipid

A large amount of evidence has demonstrated the benefits of lipid-lowering therapy on the primary prevention of cardiovascular diseases because isolated time point of lipid is limited for reflecting the overall atherosclerotic burden. Cardiovascular risk factors should evaluate two components: The magnitude and duration of sustained risk factor elevation. Therefore, cardiologists proposed the concepts of annual time-averaged exposure, cumulative exposure, visit-to-visit variability, and trajectories [21,22,23]. These relationships between risk factors and time-series changes measured from different perspectives reflect the real world from every aspect. Maintaining optimal lipid levels is necessary to achieve ideal cardiovascular health. According to the Third Report of the National Cholesterol Education Program (NCEP) Expert Panel on Detection, Evaluation, and Treatment of High Blood Cholesterol in Adults final report (NCEP-ATP III) and the latest 2016 Chinese guidelines for the prevention and treatment of dyslipidemia in adults, values of lipid and lipoprotein were classified as acceptable, borderline high, and high levels [24,25]. It is well-known that apolipoprotein B (apo B)-containing lipoproteins transport cholesterol and other lipids throughout the body and play a central role in the initiation and progression of atherosclerosis [26]. In this study, borderline-elevated baseline lipid levels were independently associated with CAS progression. Over time, borderline-elevated baseline lipid levels (TC, TG, and LDL-C) with stable/increasing trajectories leading to more retention of apo B-containing lipoprotein particles in the subintimal arterial wall provokes a complex, maladaptive inflammatory process that leads to the initiation and progression of an atheroma. This trend can be seen especially in the middle-aged population, resulting in quantitative to qualitative change under cumulative cholesterol exposure.

### 4.2. Comparisons with Existing Studies

According to the systematic review, the substantial global burden of CAS (including cIMT and carotid plaque) is estimated to be 21–28% [27]. Previous trajectory modeling publications often focused on various cardiovascular risk factors and incidents. Only some studies have focused on the relationship between subclinical outcome (CAS) and the risk factor trajectories, such as cardiovascular health, BMI, alcohol consumption, physical activity, etc. To date, our study is the only one that examines the relationship between lipid trajectories and CAS progression. In lipid trajectories, present studies often investigated only the development of specific lipid components, neglecting the involvement of trajectory characteristics. The trajectory of these lipids was only considered by Dayimu and Fatemeh [12,28]. Although both studies emphasized incident cardiovascular disease, such as our study, Dayimu et al. showed three distinct trajectory classes of lipid profiles (U-shape class, progressing, and inverse U-shape). In addition, Fatemeh et al. showed seven different multi-trajectory groups aged 45–84. Considering the practicality of clinical application, we combined baseline and over the longitudinal course for assessment [29].

Interestingly, we found no prominent relationship between cholesterol trajectories (TC, LDL-C, and HDL-C) and CAS progression without assessing baseline levels. This may be caused by the heterogeneity in the risk factors across trajectories. This indicates the inaccuracy of the assessment of outcomes merely from trajectory characteristics without considering the underlying risk profile. In the present study, we have considered the impact of both the baseline level and trajectory of risk factors on CVD progression, suggesting that a constantly controlled risk factor profile is essential for a suppressed disease progression.

### 4.3. Diet and Nutrition as Drivers of Lipid Levels

Diet and lifestyle play a significant role in dyslipidemia. Dietary modification is one of the most important measures for preventing and treating dyslipidemia. Several studies demonstrate an inverse relationship between lipid levels and certain food groups (such as fish, fruits, and whole grains) or specific diet patterns (such as the MedDiet or DASH diet [30]). Furthermore, an increasing number of studies suggest that eating patterns are closely related to the metabolism of lipids in both animals and humans. The habit of eating at night and irregular eating patterns have been found to be positively associated with cardiovascular disease in the general population [31,32]. The current study was unable to investigate the relationship between lifestyle and CAS progression since there were no questionnaires related to lifestyle and diet. Moreover, further research is needed to clarify this issue.

### 4.4. Strength and Limitations

The strengths of this study include the large population longitudinal cohort with the repeated measurements of lipid profiles. The method was well-designed using LCTM, which enabled us to obtain detailed insight into the longitudinal lipid trajectories. However, several limitations of the study should also be stated, as follows. First, our data were derived from routine health checkups in a single center in China. The results of the current study may not be generalizable to other populations without these demographic characteristics. Second, participants were collected from the general health checkups, lacking sufficient medical history and medication information. It is possible to maintain a normal level of lipid in patients by taking lipid-lowering medications. The results for lipid trajectory and CAS progression may be underestimations of the true associations in general health checkups. Dyslipidemia patients in China, even when combined with cardiovascular disease, are less likely to take medication, have poor control rates, and are less likely to take medication on long-term stable levels [2]. In general, the proportion associated with long-term lipid management should be insignificant. Third, the identification of qualitative CAS progression was based on the ultrasound reports from the medical records. Moreover, the detection kit and biological variability of the lipid profiles would impact the association estimation.

## 5. Conclusions

Borderline elevated baseline lipid (TC, TG, and LDL-C) with stable and elevated-increasing trajectories were associated with CAS progression. Moreover, long-term strategies for low-level lipids are beneficial for ASCVD management.

## Figures and Tables

**Figure 1 nutrients-14-03243-f001:**
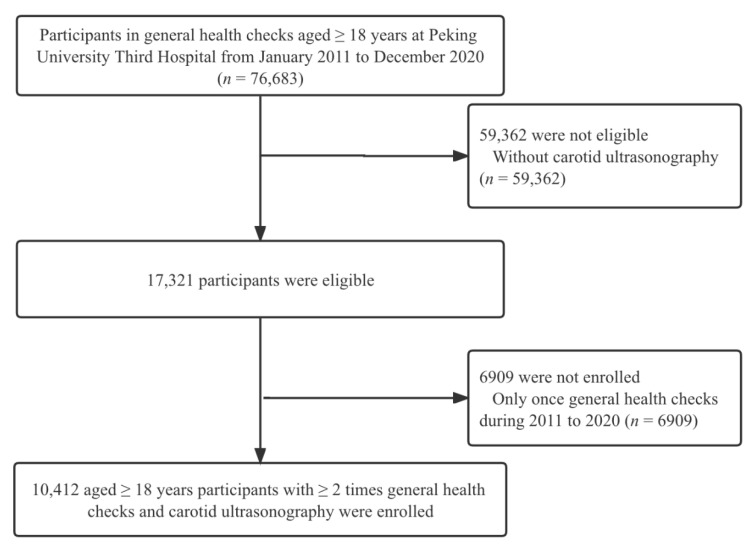
Study participants flow diagram. CAS: Carotid atherosclerosis.

**Figure 2 nutrients-14-03243-f002:**
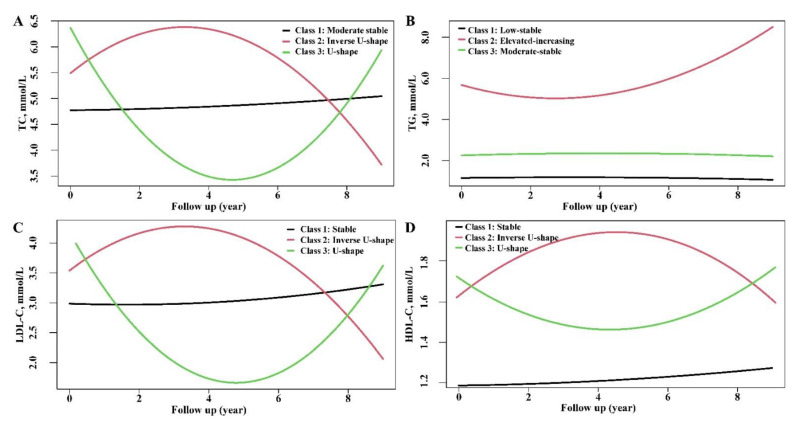
Trajectories of total cholesterol (**A**), triglycerides (**B**), low-density lipoprotein cholesterol, (**C**), and high-density lipoprotein cholesterol (**D**). Abbreviations: TC: Total cholesterol; TG: Triglyceride; LDL-C: Low-density lipoprotein cholesterol; HDL-C: High-density lipoprotein cholesterol.

**Figure 3 nutrients-14-03243-f003:**
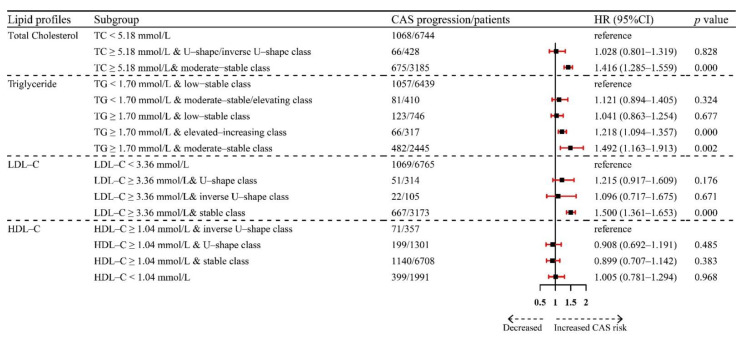
Forest plot showing the association between CAS progression and lipid trajectories from baseline. Abbreviations: CAS: Carotid atherosclerosis progression; HR: Hazard ratio; CI: Confidence interval; TC: Total cholesterol; TG: Triglyceride; LDL-C: Low-density lipoprotein cholesterol; HDL-C: High-density lipoprotein cholesterol.

**Table 1 nutrients-14-03243-t001:** Baseline characteristics of study participants.

Variables	Total (*n* = 10,412)	Normal (*n* = 5583)	Carotid Intima-Media Thickness (*n* = 1373)	Carotid Artery Plaque (*n* = 3456)	*p* for Trend
Age, years	50 (40–61)	42 (34–50)	54 (48–61)	63 (55–73)	0.000
Male, *n* (%)	5421 (52.1)	2712 (48.6)	733 (53.4)	1976 (57.2)	0.000
BMI, kg/m^2^	24.42 (22.21–26.71)	23.87 (23.60–26.21)	24.85 (22.76–26.95)	25.07 (23.11–27.23)	0.000
Hypertension, *n* (%)	1584 (15.2)	343 (6.1)	214 (15.6)	1027 (29.7)	0.000
Diabetes, *n* (%)	505 (4.9)	95 (1.7)	51 (3.7)	359 (10.4)	0.000
SBP, mmHg	127 (115–140)	121 (111–132)	129 (117–140)	137 (124−149)	0.000
DBP, mmHg	77 (69−84)	74 (67−82)	79 (71−86)	80 (72−87)	0.000
FBG, mmol/L	5.20 (4.80−5.70)	5.00 (4.70−5.40)	5.20 (4.90−5.70)	5.50 (5.10−6.30)	0.000
TC, mmol/L	4.82 (4.21−5.43)	4.70 (4.16−5.31)	5.01 (4.41−5.58)	4.94 (4.24−5.58)	0.000
TG, mmol/L	1.35 (0.96−1.96)	1.25 (0.89−1.85)	1.45 (1.04−2.02)	1.49 (1.08−2.11)	0.000
HDL-C, mmol/L	1.26 (1.08–1.49)	1.27 (1.08–1.49)	1.28 (1.10–1.50)	1.25 (1.07–1.47)	0.009
LDL-C, mmol/L	3.03 (2.49–3.58)	2.95 (2.46–3.47)	3.20 (2.68–3.74)	3.10 (2.46–3.72)	0.000

Abbreviations: BMI: Body mass index; SBP: Systolic blood pressure; DBP: Diastolic blood pressure; FBG: Fasting blood glucose; TC: Total cholesterol; TG: Triglyceride; LDL-C: Low-density lipoprotein cholesterol; HDL-C: High-density lipoprotein cholesterol.

**Table 2 nutrients-14-03243-t002:** Baseline demographic and clinical characteristics according to lipid trajectories defined based on latent class trajectory models.

Variables	Total Cholesterol			Triglyceride			LDL-C			HDL-C		
	Class 1: Moderate-Stable (*n* = 9917)	Class 2: Inverse U-Shape (*n* = 222)	Class 3: U-Shape (*n* = 257)	Class 1: Low-Stable (*n* = 7216)	Class 2: Elevated-Increasing (*n* = 334)	Class 3: Moderate-Stable (*n* = 2846)	Class 1: Stable (*n* = 9909)	Class 2: Inverse U-Shape (*n* = 148)	Class 3: U-Shape (*n* = 339)	Class 1: Stable (*n* = 8731)	Class 2: Inverse U-Shape (*n* = 359)	Class 3: U-Shape (*n* = 1306)
Age, years	**50 (39–61)**	**54 (48–64)**	**58 (53–65)**	**49 (38–61)**	51 (42–58)	**53 (43–62)**	**50 (39–61)**	**54 (47–64)**	**58 (52–65)**	**50 (39–61)**	52 (42–62)	**53 (44–63)**
Male	**5227 (52.7)**	75 (33.8)	108 (42.0)	**3380 (46.8)**	**249 (74.6)**	**1781 (62.6)**	**5209 (52.6)**	58 (39.2)	143 (42.2)	**3685 (42.2)**	80 (22.3)	284 (21.7)
BMI, kg/m^2^	**24.39 (22.19–26.67)**	24.55 (22.47–27.17)	**25.36 (22.91–27.67)**	**23.71 (21.58–25.91)**	26.06 (24.24–27.98)	25.84 (23.95–28.06)	**24.39 (22.18–26.65)**	25.19 (23.06–27.26)	25.33 (22.85–27.73)	**24.80 (22.68–27.04)**	22.19 (20.31–24.27)	22.23 (20.41–24.31)
Hypertension	1482 (14.9)	34 (15.3)	**67 (26.1)**	**938 (13.0)**	80 (24.0)	565 (19.9)	1475 (14.9)	20 (13.5)	**88 (26.0)**	**1375 (15.7)**	47 (13.1)	**161 (12.3)**
Diabetes	472 (4.8)	9 (4.1)	**24 (9.3)**	320 (4.4)	**37 (11.1)**	148 (5.2)	471 (4.8)	5 (3.4)	29 (8.6)	**446 (5.1)**	14 (3.9)	**45 (3.4)**
SBP, mmHg	**126 (115–139)**	**130 (117–142)**	**137 (124–148)**	**124 (113–138)**	**135 (123–147)**	**131 (121–144)**	**126 (115–139)**	131 (119–144)	135 (122–146)	**127 (116–140)**	122 (110–134)	123 (111–137)
DBP, mmHg	77 (69–84)	78 (70–87)	**81 (74–88)**	**75 (68–83)**	**84 (77–91)**	**80 (73–87)**	**77 (69–84)**	77 (70–86)	**81 (73–88)**	**78 (70–85)**	74 (66–82)	74 (67–82)
Baseline biochemical profiles												
TC, mmol/L	**4.77 (4.19–5.36)**	**6.32 (5.55–7.04)**	**6.56 (5.98–7.23)**	**4.69 (4.11–5.31)**	**5.34 (4.55–6.15)**	**5.10 (4.49–5.72)**	**4.77 (4.19–5.36)**	**6.10 (5.09–6.90)**	**6.32 (5.70–6.84)**	**4.75 (4.16–5.38)**	**4.94 (4.30–5.48)**	**5.18 (4.63–5.77)**
TG, mmol/L	**1.33 (0.95–1.93)**	**1.67 (1.24–2.48)**	**2.05 (1.43–2.96)**	**1.10 (0.84–1.42)**	**5.17 (3.93–6.95)**	**2.29 (1.92–2.82)**	**1.33 (0.95–1.94)**	1.78 (1.26–2.35)	1.72 (1.28–2.33)	**1.46 (1.04–2.09)**	**1.00 (0.78–1.36)**	**0.92 (0.72–1.21)**
HDL-C, mmol/L	**1.26 (1.08–1.49)**	**1.36 (1.16–1.58)**	1.27 (1.11–1.52)	**1.35 (1.16–1.57)**	**0.99 (0.87–1.12)**	**1.11 (0.98–1.27)**	1.26 (1.08–1.49)	1.26 (1.05–1.46)	1.28 (1.15–1.49)	**1.21 (1.05–1.38)**	**1.63 (1.49–1.77)**	**1.80 (1.71–1.96)**
LDL-C, mmol/L	**2.99 (2.47–3.52)**	**4.15 (3.44–4.87)**	**4.37 (3.82–4.84)**	**2.95 (2.43–3.50)**	**2.64 (2.02–3.30)**	**3.27 (2.72–3.80)**	**2.99 (2.46–3.51)**	**4.14 (2.99–5.01)**	**4.37 (3.91–4.82)**	**3.05 (2.52–3.60)**	2.86 (2.32–3.41)	2.94 (2.43–3.51)
FBG, mmol/L	**5.2 (4.8–5.7)**	**5.4 (5.0–6.0)**	**5.6 (5.2–6.4)**	**5.1 (4.8–5.5)**	**5.6 (5.1–6.8)**	**5.4 (5.0–6.0)**	**5.2 (4.8–5.7)**	5.3 (5.0–6.1)	5.5 (5.1–6.2)	**5.2 (4.9–5.7)**	5.1 (4.7–5.5)	5.1 (4.7–5.5)

Bold values denote significant difference between the groups. Significance values have been adjusted by the Bonferroni method. Abbreviations: BMI: Body mass index; SBP: Systolic blood pressure; DBP: Diastolic blood pressure; FBG: Fasting blood glucose; TC: Total cholesterol; TG: Triglyceride; LDL-C: Low-density lipo-protein cholesterol; HDL-C: High-density lipoprotein cholesterol.

**Table 3 nutrients-14-03243-t003:** Hazard ratios (95% confidence intervals) of CAS progression by baseline level and trajectory class of each lipid profile.

Baseline Lipid Profiles and Longitudinal Trajectory	*n*	CAS Progression	Unadjusted		Model 1		Model 2	
			Crude HR (95% CI)	*p* Value	Adjusted HR (95% CI)	*p* Value	Adjusted HR (95% CI)	*p* Value
TC								
Class 1: Moderate-stable	9917	1738	Reference		Reference		Reference	
Class 2: Inverse U-shape	222	40	0.905 (0.661–1.239)	0.534	0.876 (0.640–1.201)	0.411	0.849 (0.617–1.169)	0.317
Class 3: U-shape	257	36	0.910 (0.654–1.267)	0.577	0.835 (0.599–1.164)	0.287	0.824 (0.585–1.159)	0.266
TG								
Class 1: Low-stable	7216	1185	Reference		Reference		Reference	
Class 2: Elevated-increasing	334	69	**1.497 (1.174–1.908)**	**0.000**	**1.442 (1.129–1.841)**	**0.003**	**1.417 (1.101–1.824)**	**0.007**
Class 3: Moderate-stable	2846	560	**1.192 (1.078–1.318)**	**0.000**	**1.144 (1.034–1.266)**	**0.009**	1.084 (0.974–1.207)	0.139
LDL-C								
Class 1: Stable	9909	1730	Reference		Reference		Reference	
Class 2: U-shape	339	57	1.076 (0.826–1.402)	0.585	0.993 (0.761–1.295)	0.960	0.978 (0.744–1.286)	0.874
Class 3: Inverse U-shape	148	27	0.832 (0.568–1.219)	0.345	0.788 (0.538–1.155)	0.221	0.759 (0.514–1.121)	0.165
HDL-C								
Class 1: Stable	8731	1544	Reference		Reference		Reference	
Class 2: Inverse U-shape	359	71	1.079 (0.851–1.369)	0.530	1.122 (0.883–1.427)	0.346	1.217 (0.953–1.554)	0.116
Class 3: U-shape	1306	199	0.983 (0.848–1.140)	0.819	1.023 (0.879–1.191)	0.766	1.089 (0.930–1.275)	0.292
Baseline borderline high levels								
TC ≥ 5.18 mmol/L	3613	741	**1.371 (1.248–1.506)**	**0.000**	**1.331 (1.210–1.464)**	**0.000**	**1.316 (1.194–1.451)**	**0.000**
TG ≥ 1.70 mmol/L	3508	671	**1.193 (1.084–1.312)**	**0.000**	**1.136 (1.031–1.251)**	**0.010**	1.061 (0.957–1.176)	0.263
LDL-C ≥ 3.36 mmol/L	3592	740	**1.461 (1.330–1.605)**	**0.000**	**1.417 (1.289–1.557)**	**0.000**	**1.382 (1.254–1.523)**	**0.000**
HDL-C < 1.04 mmol/L	1991	399	1.107 (0.994–1.242)	0.063	1.067 (0.951–1.197)	0.271	1.021 (0.998–1.217)	0.739

Bold values denote statistical significance. Model 1: Adjusted for age and male; Model 2: Adjusted for age, male, BMI, hypertension, and diabetes. Abbreviations: CAS: Carotid atherosclerosis progression; HR: Hazard ratio; TC: Total cholesterol; TG: Triglyceride; LDL-C: Low-density lipoprotein cholesterol; HDL-C: High-density lipoprotein cholesterol.

## Data Availability

Not applicable.

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
