# Peer review of "Trajectories of Lipid Profile and Risk of Carotid Atherosclerosis Progression: A Longitudinal Cohort Study"

_nutrients, 2022, doi:10.3390/nu14153243_

Round 1

Reviewer 1 Report

This paper reports on a large longitudinal cohort study with over 10,000 participants with the specific aim to look at the association between changes in lipid profiles and carotid atherosclerosis (CAS) progression.  Latent Class Trajectory Modelling (LCTM) is used to identify trajectories of the lipid profiles over 8 years follow up.  The use of LCTM is interesting, and given that the method is reasonably new this is likely to provide extra insight into what has typically been done previously.  I am able to comment on the statistical methods used and I hope the comments/suggestions below are useful.

It is stated that continuous measures are summarized using median and IQRs, or means and SDs as guided by the Kolmogorov-Smirnov tests.  Presumably this specifically is to test for deviation away from normality?  It would be good to make this clear.  However, as far as I can tell all data is summarized using medians and IQRs.  This is not surprising given the sample size, since the test is very high powered to detect even mild deviations away from normality (but even then, I think a mean and SD is still appropriate even when normality is violated and the data is still reasonably symmetric).  It might be easier just to state that medians and IQR are reported without mention of the need for a test of normality.

Is the P for trend in table simply a p-value for a test of any differences between the groups?  Or does trend mean specifically testing whether the carotid artery plaque group is greater than the cIMT group which in turn is greater than the normal group?  

On that, the median and IQRs for some of the measures appear very similar and point to perhaps no meaningful differences.  E.g. HDL, where the P-value was small.  Due to the large sample sizes it would be to be clearer on statistical versus clinical significance, and therefore which of the statistically significant findings are actually indicative of some meaningful difference.  

The trajectory classes analysis was interesting.  In each case three classes were identified.  Was the analysis requested to find a maximum of three?  Or does the model always seek three classes, or perhaps a coincidence that three were identified for each?  Also, more details would be useful too (e.g. random effect, as an intercept etc.?).

Also, given that these classes are themselves estimated from the data, is there any supporting statistics that can accompany these to indicate how well the trajectory described the individuals in the classes?  E.g., is someone was described well by any of these (and there must be some), what class do they belong?

Reviewer 2 Report

In this article, the authors aimed to identify different trajectories of lipid profiles and investigate the association of lipid trajectories with Carotid atherosclerosis (CAS) progression in a large, longitudinal cohort of the Chinese population (a total of 10,412 participants). The results showed that borderline elevated base-line lipid (TC, TG, and LDL-C) with stable and elevated-increasing trajectories were associated with CAS progression. Some comments are as follows:   

1.      Although that dyslipidemia is a well-established traditional risk factor for atherosclerosis, insulin resistance (IR) plays an important role in CAS progression. Please show the effects of IR on CAS in this large, longitudinal cohort.

2.      Besides, atherogenic index (AI), which corresponds to the ratio of TC/HDL-C, is related to CAS progression. It would be interesting to enroll AI in this analysis.  

3.      Given a strong effect of the used statins or others lipid-lowering agents on lipid profiles, the authors should illustrate the related data and have some discussion.

4.      Age is an important predisposing marker in CAS. It would be better to stratify the age levels in an association of lipid trajectories with Carotid atherosclerosis (CAS) progression.

Some typing errors were noted.
